# Research on Temperature Compensation of Multi-Channel Pressure Scanner Based on an Improved Cuckoo Search Optimizing a BP Neural Network

**DOI:** 10.3390/mi13081351

**Published:** 2022-08-19

**Authors:** Huan Wang, Qinghua Zeng, Zongyu Zhang, Hongfu Wang

**Affiliations:** School of Aeronautics and Astronautics, Sun Yat-sen University, Shenzhen 518107, China

**Keywords:** multi-channel, pressure scanner, temperature compensation, chaotic simplex algorithm, cuckoo search algorithm, BP neural network

## Abstract

A multi-channel pressure scanner is an essential tool for measuring and acquiring various pressure parameters in aerospace applications. It is important to note, however, that the pressure sensor of each of these channels will drift significantly with the increase in the temperature range of the pressure measurement, and the output voltage of each of these channels will show nonlinear characteristics, which will constrain the improvements in the accuracy of the measurement. In the regression fitting process, it is difficult to fit nonlinear data with the traditional least-squares method, which leaves pressure measurement accuracy unsatisfactory. A temperature compensation method based on an improved cuckoo search optimizing a BP neural network for a multi-channel pressure scanner is proposed in this paper to improve pressure measurement accuracy in a wide temperature range. Using the chaotic simplex algorithm, we first improved the cuckoo search algorithm, then optimized the connection weights and thresholds of the BP neural network, and finally constructed an experimental calibration system to investigate the temperature compensation of the multi-channel pressure scanning valves in the −40 °C to 60 °C temperature range. The compensation test results show that the algorithm has a better compensation effect and is more suitable for the temperature compensation of multi-channel pressure scanners than the traditional least-squares method and the standard RBF and BP neural networks. The maximum full-scale error of all 32 channels is 0.02% FS (full-scale error) and below, which realizes its high-accuracy multi-point pressure measurement in a wide temperature range.

## 1. Introduction

Pressure scanners are developed from multi-point pressure measurement systems [1,2], primarily to measure and monitor multi-point pressure simultaneously. In their early stages, the devices were developed to measure distributed forces in wind tunnels and fluids for military and aviation development [3]. They are now also used in engine tests, impeller and pressurizer tests, and industrial flow fields [4]. As a result of their high-performance indicators, such as their degree of integration, pressure measurement accuracy, and real-time calibration, multi-channel pressure scanners are always considered to be at the forefront of the measuring industry.

As aerospace technology develops, many high-performance pressure scanners are used for aircraft and engine measurements. However, the original measurement range and accuracy cannot fully satisfy the current model requirements. In response to this phenomenon, Pressure System Inc. (PSI) released the PSI9000 series of pressure scanners. The main parameters of the instrument are measurement range: up to 850 psi, accuracy: ±0.05% FS, and temperature compensation range: 0 to 70 °C [5]. The DSA3200 series pressure scanner has been introduced by Scanivalve Corp, Liberty Lake, WA, USA. The pressure range is 0° to 72 °C with automatic temperature compensation, and the accuracy is ±0.05% FS [6]. With a maximum pressure range of 300 PSI and an accuracy of ±0.10% FS over the temperature compensation range of −55 to 125 °C, Kulite Semiconductor Products, Inc., Leonia, NJ, USA has successfully developed the KMPS series of pressure scanners for the flight-testing market [7]. Currently, these high-precision multi-channel pressure scanners are used in aerospace for testing purposes. In spite of this, they are unsuitable for large-scale deployment and use due to their fragility in extreme environments, high cost, and long maintenance cycles.

Several research units have developed and designed pressure scanners around performance indicators in response to this problem. Among them, achieving high accuracy measurements of multi-point pressures under high and low temperatures, as well as positive and negative pressure conditions, has been the main focus of scholars’ research. In an ultra-small volume, the pressure scanner integrates multiple pressure channels (each pressure channel contains a silicon piezoresistive pressure sensor) and multiple temperature channels, enabling simultaneous multi-channel pressure and temperature measurements. The silicon piezoresistive pressure sensor is one of the typical applications based on MEMS technology, which is a microelectromechanical sensor generated on a silicon wafer. It uses MEMS technology to directly inscribe four high-precision semiconductor strain gauges on its surface where the stress is maximum and converts the physical quantity of pressure into an electrical signal output through Wheatstone bridge measurement, thus achieving a sensitive output of force. The advantages of MEMS pressure sensors over conventional pressure sensors are: high measurement accuracy, low power consumption, and low cost.

It is worth noting that the MEMS silicon pressure sensor of each channel inside the pressure scanner will drift with temperature change when it is operating. The root cause of this is that the sensitivity coefficient and the value of the diffusion resistance change with temperature, which lowers the accuracy when measuring pressure. Temperature drift can be compensated for in two ways: with hardware and software [8]. Temperature-sensitive elements are used in hardware compensation to offset the effects of temperature drift on the pressure sensor measurements [9,10,11]. With multi-channel pressure scanners requiring high system integration and an extensive measurement range, this method can be challenging to implement and inaccurate. By using the appropriate temperature compensation algorithm, software compensation corrects temperature drift errors [12,13]. As previously applied to pressure scanners, traditional software compensation algorithms consist mainly of polynomial fitting, surface fitting with the least-squares method, and curve interpolation [14]. However, these conventional algorithms are limited by the demand for pressure measurement accuracy [15,16,17]. In particular, determining a stable solution for the equation when using the least-squares method for temperature compensation will create a pathological problem if the order of the fitted expression is high. The interpolation method can be used to complete temperature compensation, but the high interpolation can cause the Runge phenomenon; on the other hand, it is difficult to guarantee the effect of interpolating without determining the boundary.

Since the development of intelligent optimization algorithm theory, neural network algorithms have performed well in the field of sensor temperature compensation due to their nonlinear solid mapping abilities and generalization abilities [18,19,20]. However, standard neural networks are not suitable for handling all types of temperature compensation problems and are often combined with machine learning algorithms to better exploit the advantages of neural networks. The literature [21] shows that the BP neural network temperature compensation method is effective, reliable, and robust, and can be extended to similar sensors. One study [22] shows that artificial neural networks can be applied for temperature compensation in 3D stress sensors. Another study [23] illustrates the application of genetic neural networks to sensor array compensation. A temperature compensation model using an improved particle swarm optimization RBF neural network was developed [24], which showed a good compensation effect for silicon pressure sensors. A neural network temperature compensation system is proposed in [25] to improve the accuracy of the pressure measurement system to 0.2% FS. A genetic wavelet neural network compensation model is proposed in [26] to achieve the reduction of temperature nonlinearity.

By combining neural network and machine learning algorithms as models, this paper proposes a method for improving the pressure measurement accuracy of multi-channel pressure scanners by using an improved cuckoo search algorithm (CS) to optimize BP neural networks for its problems and shortcomings in the actual calibration process and temperature compensation. A global optimization capability of the improved cuckoo search algorithm is combined with the initial weights and thresholds of the BP neural network to reduce the dependence of the algorithm on the initial values, which can easily fall into the local extremes and be affected by the threshold values. Lastly, we combine the algorithm with the 32-channel experimental data of the pressure scanner, and the experimental results show that the method can effectively improve the fitting accuracy compared with the traditional numerical calculation method and the standard neural network algorithm, and has better generalization ability.

The following is a list of the paper’s main contributions:(1)A multi-channel temperature compensation model for pressure scanners is proposed;(2)Introducing the cuckoo algorithm into the field of multi-channel pressure sensor temperature compensation and improving the cuckoo algorithm by proposing a multi-channel high-precision compensation algorithm combined with neural networks;(3)The establishment of an experimental calibration system for multi-channel pressure scanners;(4)Our analysis and comparison of compensation results from different algorithms are combined with the error evaluation index, and the CS-BPNN algorithm is then applied to the compensation of the 32-channel pressure sensor of the pressure scanner.

## 2. Temperature Compensation Algorithm and Calibration Experimental System

### 2.1. Improved CS-BPNN Temperature Compensation Algorithm

The temperature compensation algorithm was designed according to the multi-channel pressure scanner characteristics of pressure measurement, and its compensation principle can be seen in Figure 1. Figure 1 illustrates the process of compensating the pressure sensors in each channel of the 32-channel pressure scanner. For example, the voltage signal output from the pressure sensor in channel 1 is $U_{P1}$, and the output corresponding to the temperature is $T_{1}$. These two components are used as inputs to the neural network model algorithm, and each set of inputs will correspond to a predicted pressure value $P_{1}$ of the model output. In order to determine whether the compensation is effective, the error index between the predicted and calibrated pressure values are compared. As a final step, we compensate the pressure sensors of 32 channels in turn.

#### 2.1.1. BP Neural Network

According to the error back-propagation algorithm, BP neural networks are multilayered feedforward neural networks [27,28]. With powerful nonlinear mapping capabilities, they are easy to reverse and therefore it is straightforward to find the relationship between input and output. Figure 2 below illustrates the basic structure of a practical 3-layer BP neural network.

We will assume that the input layer is composed of n neurons, the hidden layer is composed of q neurons, and the output layer is composed of m neurons. For the j node of the hidden layer, define $net_{j}$ as the input and $c_{j}$ as the output. Using the output layer, the input of the k node is $net_{k}$, followed by the output of $c_{k}$. The threshold values for the hidden layer and output layer are $\theta_{j}$ and $\theta_{k}$, respectively. Connection weights $w_{ij}$ between the input layer and hidden layer, and $w_{jk}$ between the hidden layer and output layer. Assuming $X={\left[x_{1},x_{2},\cdot \cdot \cdot x_{n}\right]}^{T}\;$ is the input vector, $Y={\left[y_{1},y_{2},\cdot \cdot \cdot y_{m}\right]}^{T}$ is the desired output vector. From the BP neural network feedforward calculation, the input of the hidden layer’s j node is
(1)$$net_{j}=\overset{n}{\underset{i=1}{\sum }}w_{ij}c_{i}$$

At the hidden layer’s j node, the output is
(2)$$c_{j}=f\left(net_{j}\right)=\frac{1}{1+e^{-\left(net_{j}-\theta_{j}\right)}}$$

The k node of the output layer receives the following input:(3)$$net_{k}=\overset{q}{\underset{j=1}{\sum }}w_{jk}c_{j}$$

The output of the k node of the corresponding output layer (the predicted output of the neural network) is:(4)$$c_{k}=f\left(net_{k}\right)$$

An average cost function of the system is introduced when there is an error between the predicted and desired output of the neural network:(5)$$E=\frac{1}{2}\overset{m}{\underset{k=1}{\sum }}{\left(y_{k}-c_{k}\right)}^{2}$$

To minimize the cost function E, the connection weights should be adjusted in accordance with the inverse of the gradient change, so the predicted output of the neural network is close to the desired value. The weights of the output layer are modified as follows:(6)$$\Delta w_{jk}=-\eta \frac{\partial E}{\partial w_{jk}}=-\eta \frac{\partial E}{\partial net_{k}}\frac{\partial net_{k}}{\partial w_{jk}}=-\eta \frac{\partial E}{\partial c_{k}}\frac{\partial c_{k}}{\partial net_{k}}\frac{\partial net_{k}}{\partial w_{jk}}$$

In Equation (6), η is the learning rate and generally η>0. Additionally, because
(7)$$\frac{\partial E}{\partial c_{k}}=-\left(y_{k}-c_{k}\right)$$
(8)$$\frac{\partial c_{k}}{\partial net_{k}}=\frac{\partial f\left(net_{k}\right)}{\partial net_{k}}=f^{\prime }\left(net_{k}\right)$$

The weight adjustment formula for the output layer is as follows:(9)$$\Delta w_{jk}=\eta \left(y_{k}-c_{k}\right)c_{k}\left(1-c_{k}\right)c_{j}$$

In the same way, we can obtain the formula for adjusting the hidden layer’s weight:(10)$$\Delta w_{ij}=\eta c_{j}\left(1-c_{j}\right)\left[\overset{m}{\underset{k=1}{\sum }}c_{k}\left(1-c_{k}\right)\left(y_{k}-c_{k}\right)w_{jk}\right]c_{i}$$

The transfer functions for the hidden layer and output layer neurons are *tansig* and *purelin*, respectively, and they are calculated as below. Moreover, the training method is *trainlm* (Levenberg–Marquardt algorithm). The fitness function is the squared sum of predicted and expected values.
(11)$$f\left(net_{j}\right)=\frac{2}{1+e^{-2net_{j}}}-1$$
(12)$$f\left(net_{k}\right)=net_{k}$$

#### 2.1.2. Improved Cuckoo Search Algorithm

In 2009, Yang X.S. and Deb S. proposed a nature-inspired cuckoo search algorithm [29]. It has several advantages, including fewer parameters, an efficient global search, fast convergence, better generality, and robustness. In practice, when dealing with optimization of complex problems, it is not necessary to adjust a large number of parameters but in essence only to match a probability parameter $p_{a}$. In recent studies, cuckoo search has also been more accurate than other heuristics (e.g., particle swarm optimization algorithms, etc.) [30,31].

In general, the following three idealized rules can be used to describe the optimization process of the cuckoo search algorithm.

(1)Each cuckoo lays one egg at a time and places it randomly in the nest.(2)Nests with good quality eggs will be retained for the next generation.(3)The number of available parasitic nests is fixed, and the host has a probability of finding an egg placed by a cuckoo. In this case, the host can either discard the egg or create a new nest.

Each egg in the nest represents a solution and each cuckoo can only lay one egg to replace bad solutions with new and possibly better ones. The cuckoo search algorithm can also be extended to have multiple eggs per nest to represent a set of solutions. In this paper, we consider only the simplest case where each nest has only one egg. In other words, there is no difference between an egg, a nest, or a cuckoo, since each nest corresponds to an egg and accordingly represents a cuckoo.

Global random walk using Levy flight:(13)$${x^{\prime }}_{in}^{\left(t+1\right)}={x^{\prime }}_{in}^{\left(t\right)}+\alpha ⊕L\left(\lambda \right)$$

In the above equation, ${x^{\prime }}_{in}^{\left(t+1\right)}$ and ${x^{\prime }}_{in}^{\left(t\right)}$ are two different solutions generated by random walk, α>0 is the step scale factor, and ⊕ denotes point-to-point multiplication. (14)$$L\left(\lambda \right)=\frac{1}{\pi }\left[\lambda \Gamma \left(\lambda \right)\sin(\frac{\pi \lambda }{2})\right]$$

In the above equation, $\Gamma \left(\lambda \right)$ is the gamma function, also known as the Euler quadratic integral, which is a function of the real and complex numbers after factorial multiplication, and where the gamma function in the real number domain is defined by
(15)$$\Gamma \left(x\right)=\int_{0}^{+\infty }t^{x-1}e^{-t}dt$$

At this point, the basic steps of the CS algorithm are given as follows:

Step 1 Initialize the basic parameters and randomly generate N initial bird-nesting positions.

Step 2 Calculate the test values of each nest location and determine the current best test value and the best nest location.

Step 3 Update the bird nest locations using Equations (13) and (14) to obtain a new set of bird nest locations.

Step 4 Calculate the test values of the updated bird nest positions, compare the positions of the previous generation of bird nests, and the nest with the better test value position goes to the next step.

Step 5 Generate a random number $r\in \left(0,1\right)$ to compare with the probability $p_{a}$ that the bird nest is found. The nest with a lower probability of being found is kept, while the nest with a higher probability of being found is randomly changed to get a new set of nest locations.

Step 6 Determine whether the algorithm satisfies the termination condition; if it does, the global optimal solution is output and the algorithm ends; otherwise, return to Step 3.

The standard cuckoo algorithm, however, performs poorly in the late iterations and has slow convergence. Due to this, researchers are more likely to embed chaotic optimization methods as a local search technique into heuristic search algorithms to obtain hybrid algorithms with better search performance, such as the chaotic ant colony algorithm [32], chaotic particle swarm algorithm [33], chaotic firefly algorithm [34], and chaotic differential evolution algorithm [35], etc. Nowadays, to accelerate the convergence of heuristic algorithms, many heuristic algorithms use the simplex method due to its simplicity and local search ability [36]. Here, the simplex method is not described in detail. An algorithmic approach to improving the cuckoo search algorithm is presented in this paper using chaotic simplex algorithms [37]. In chaotic optimization, chaotic mapping is used to generate chaotic sequences. Due to the randomness, ergodicity, and acyclic nature of the chaotic sequences, chaos optimization has a higher efficiency than other stochastic optimization algorithms [38].

The chaotic simplex algorithm for optimizing the cuckoo search includes the following steps:

(a) By using the simplex method, the poorer solution in the current nest location is removed.

(b) Calculate the test value of the best-positioned nest ${x^{\prime }}_{best}$ in the current population.

(c) In the vicinity of ${x^{\prime }}_{best}$, the chaos mapping sequence is constructed using logistic chaos mapping. As you can see, the formula is as follows:(16)$$p=\frac{\left(x^{\prime }-{x^{\prime }}_{min}\right)}{\left({x^{\prime }}_{max}-{x^{\prime }}_{min}\right)}$$(17)$$p_{s+1}=\mu p_{s}\left(1-p_{s}\right)$$

In this case, ${x^{\prime }}_{min}$ and ${x^{\prime }}_{max}$ represent the minimum and maximum nest positions, respectively. In this paper, we take μ=3, the chaos coefficient, which is in the range $\mu \in \left[0,4\right]$. $p_{s}$ is the s chaos number, and iterations are denoted by s. In $\left(0,1\right)$, chaotic sequences are generated by initial conditions $p_{0}\in \left(0,1\right)$ and $p_{0}\notin \left\{0,0.25,0.5,0.75,1\right\}$.

(d) The chaotic bird’s nest location $p_{new}$ is obtained by mapping the chaotic mapping sequence $p_{s}$ back to the solution space. Following is the equation:(18)$$p_{new}={x^{\prime }}_{min}+\left({x^{\prime }}_{max}-{x^{\prime }}_{min}\right)\times p_{s}$$

(e) Compare the test value of the chaotic nest position with the current nest position and decide whether to replace the nest at the current position ${x^{\prime }}_{best}$.

#### 2.1.3. Improved CS Optimizing a BP Neural Network

The improved CS-BPNN algorithm consists of the following main ideas: First, the initial weights and thresholds of the BP neural network are assigned to the bird’s nest position of the cuckoo search algorithm; then, the chaotic simplex algorithm is embedded as a local search technique into the cuckoo search algorithm, which has the advantage of global search; then, the initial weights and thresholds of the most suitable BP neural network are quickly found by the improved cuckoo search algorithm (the minimum sum of squared errors between predicted and expected values is used as the criterion for determining the optimal initial weights and thresholds); finally, the predicted pressure value of the current pressure sensor is obtained by the BP neural network. Figure 3 shows its specific algorithm flow chart.

### 2.2. Calibration Experiment System

The calibration experimental system of the multi-channel pressure scanner consisted of software and hardware parts, and the schematic diagram is shown in Figure 4a.

In summary, it consisted of the following components: (a) BM400 air compressor; (b) SY-3000H vacuum pump; (c) ITECH IT6332B DC regulated power supply; (d) thermostatically and humidity-controlled test chamber; (e) sealing device; (f) 32-channel pressure scanner; (g) test computer; (h) ALKC400H precision digital pressure gauge; and (i) digital temperature–atmosphere pressure gauge. The flow chart in Figure 4b shows the steps of the calibration experiment.

A 32-channel pressure acquisition controller and four 8-channel pressure sensor modules were used in this experiment to construct a multi-channel pressure scanner. The overall calibration test was performed for a pressure range of 0~1.1 MPa (absolute pressure) and a temperature range of −40 °C~60 °C. In the first step, we checked the seal of the sealing device at the operating temperature and pressure. After that, we put the 32-channel pressure scanning valve into the sealing device, and we put the sealing device into the thermostatically and humidity-controlled test chamber. A 24 V DC electrical signal was drawn from the airline plug on the sealing device to power the pressure scanning valve, and the test computer was connected via an RS-422 serial interface. Through the precision digital pressure gauge, the inlet end of the sealing device was connected to the air compressor and vacuum pump, enabling the pressure change of the gas in the sealing device to be realized by electric pressurization and depressurization. Using metal bellows, the outlet of the sealing device connected to a valve and was finally returned to the temperature chamber. Lastly, a stable temperature environment of −40 °C~60 °C was simulated by the temperature chamber, and we adjusted the valves at the inlet end and outlet end so that the pressure scanner operated in different pressure and temperature environments. A test computer’s upper computer software collected voltage and temperature data from the upper 32 pressure sensors in different environments. Figure 5 shows the specific physical calibration experimental system.

## 3. Results and Discussion

### 3.1. Calibration Experiment Results

We selected 19 calibration test temperature points at 5 °C or 10 °C intervals based on the operating ambient temperature of the pressure scanner in actual use. During each temperature point, a pressure step of 100 kPa was applied for a total of 12 pressure steps. The calibration pressure $P_{V}$, the voltage output value $U_{P}$, and the temperature output value T were recorded for 32 channels of the pressure scanner at different temperatures. (The 32 channels measure the voltage signal $U_{P1}$~$U_{P32}\;$ and temperature signal $T_{1}$~$T_{32}$ at the calibrated pressure and temperature points, respectively.) The first sensor in channel one was selected for algorithm development and study, and then the sensors in the remaining channels were compensated sequentially. As shown in Table 1, the voltage output value $U_{P1}$ was recorded.

### 3.2. Analysis of Compensation Results

The analysis and processing of the calibration data were performed using Matlab2017a. Data from sensor one were divided randomly into a training set and a test set, and the training set data were normalized before processing. (The training set accounted for eighty percent of the total sample size and the test set accounted for twenty percent of the total sample size). The BP neural network has three layers: an input layer with two nodes, a hidden layer with 100 nodes, and an output layer with one node. For the improved cuckoo algorithm, the initial nest number is N=500, and the discovery probability is $p_{a}=0.25$. Chaos optimization is limited to 50 iterations, while the cuckoo algorithm is limited to 1000. In addition, the improved CS algorithm terminates when the chaotic solution outperforms the current solution or reaches the maximum number of iterations, while the BP neural network terminates when the mean square error reaches $1\times 10^{-9}$.

After the parameters of the neural network have been set, the training set data are loaded into the neural network for training, as described in Section 2.1.3. Figure 6 shows the output value of the channel one pressure sensor and the standard pressure relationship curve predicted by the improved CS-BPNN algorithm (referred to as ICS-BP in the figure).

At each stage of temperature, Figure 6 shows the pressure prediction of the training set data. There is an excellent linear relationship between predicted pressure values and calibration pressure curves, indicating that the improved CS-BPNN algorithm has high accuracy for predicting temperature and is better able to deal with the nonlinearity of the sensor output voltage. To evaluate the generalization abilities and compensation effects of different algorithms, we compared using the least-squares method, the RBF neural network, the BP neural network, and the improved CS-BPNN algorithm; the test set data were compensated for the analysis, and their experimental results are shown in Figure 7.

The 3D surface plots in Figure 7a–d illustrate the compensation effects of various algorithms. In this case, the absolute error represents the difference between the algorithm’s predicted pressure value and the actual calibrated value. The original least-squares method applied to the pressure scanning valve showed a maximum absolute error of 2.5 kPa in the low-temperature and high-pressure sections. At the same time, most of the rest fell within 1–2 kPa. Through the use of neural networks, the maximum absolute error decreased significantly. The maximum error value decreased to about 1.4 kPa and 1 kPa, respectively, when we used the RBF neural network and BP neural network. On this basis, we found that the RBF neural network had a significant error extremum point in both low- and high-voltage compensation effects. In contrast, the BP neural network was slightly better in improving the accuracy but inferior to the RBF neural network in the smoothness of compensation. Last but not least, the improved CS-BPNN algorithm proposed in this paper had a maximum absolute error of 0.2 kPa, which improved compensation accuracy by a factor of five over the standard BP neural network.

### 3.3. Evaluation of Error Indicators

In order to reflect the quantitative effect of the algorithm compensation to the greatest extent, this paper adds the full-scale error γ (F.S)—which characterizes the measurement accuracy of the pressure detection system—and the mean square error (*MSE*) and root mean square error (*RMSE*)—which reflect the average stability of the algorithm compensation—to the maximum absolute error ε (MAE) index. Here are the equations in order:(19)$$\varepsilon =\max\left|P_{r}-\overset{\wedge }{P_{r}}\right|,r=1,2,\cdot \cdot \cdot ,M$$
(20)$$\gamma =\max(\frac{\left|P_{r}-\overset{\wedge }{P_{r}}\right|}{P_{FS}})\times 100\%,r=1,2,\cdot \cdot \cdot ,M$$
(21)$$MSE=\frac{1}{M}\overset{M}{\underset{i=1}{\sum }}{\left(P_{r}-{\overset{\wedge }{P}}_{r}\right)}^{2},r=1,2,\cdot \cdot \cdot ,M$$
(22)$$RMSE=\sqrt{\frac{1}{M}\sum_{i=1}^{M}{\left(P_{r}-{\overset{\wedge }{P}}_{r}\right)}^{2}},r=1,2,\cdot \cdot \cdot ,M$$

Equations (14)–(17) show that $P_{r}$ is the predicted pressure value of the algorithm, $\overset{\wedge }{P_{r}}$ is the calibrated pressure value, M is the number of samples in the test set, and $P_{FS}$ is the full-scale pressure value. The predicted pressure values obtained by different algorithms through compensation experiments are quantified and analyzed according to the above error equation, and the results are shown in Figure 8.

It is evident from Figure 8 that the improved CS-BPNN algorithm proposed in this paper outperforms both the traditional least-squares algorithm and the standard neural network algorithm in all error metrics. The proposed algorithm achieves a full-scale error of 0.02%FS among them. This also means that the accuracy of the pressure scanning valve compensated with this algorithm is much higher than that of the current least-squares method of 0.25% FS in the same calibration environment. In addition, the algorithm increases the compensated smoothness of the BP neural network so that the mean square error value is reduced from 0.1834 to 0.0058, and the root mean square error value is reduced from 0.4283 to 0.0762.

### 3.4. Multi-Channel Test Results after Compensation

It is essential to verify the accuracy of the algorithm in this paper in predicting the pressure when the pressure scanner experiences non-calibrated points and the feasibility of multi-channel compensation. The 32-channel pressure sensors of the pressure scanner are first compensated for temperature using the improved CS-BPNN neural network algorithm. Secondly, the multi-channel pressure scanner was placed in a temperature chamber environment, and temperature points of −36 °C, −21 °C, −6 °C, 9 °C, 24 °C, 39 °C, and 54 °C were selected in a gradient manner within the temperature compensation range. Then, pressure points within the range of 0 to 1.1 MPa were selected for testing at different temperature points. A final step was then performed by determining the maximum absolute error of the pressure sensor at different pressures at the current test temperature. This is shown in Figure 9 for each channel at different pressures.

After compensation, Figure 9 shows the pressure scanner test results for 32 channels. The results demonstrate the effectiveness of the improved CS-BP algorithm for multi-channel pressure sensor temperature compensation. Meanwhile, the full-scale errors of the remaining 31 channels of the pressure scanner are all 0.02%FS and below, which indicates that the algorithm has good applicability for multi-channel pressure scanner temperature compensation. However, the pressure sensors of channels 1, 6, 12, and 17 show large errors at −6 °C, −21 °C, 39 °C, and 9 °C, respectively, reaching nearly 0.22 kPa. Although this indicator does not exceed the maximum absolute error of 0.2228 kPa for the pressure sensor test set of channel one, it still reflects the pressure sensors of these channels’ poor performance at specific temperatures. The poor performance of the pressure sensors in these channels at specific temperatures hinders further improvement of the pressure scanner accuracy.

## 4. Conclusions

Based on the problem that multi-channel pressure scanners cannot measure pressure accurately in wide temperature ranges, this study establishes a temperature compensation model and calibration experiment and proposes a high-accuracy temperature compensation method that can be applied to multi-channel pressure scanners. The compensation and test results show that, compared with the traditional least-squares method, RBF neural network, and BP neural network, the improved CS-BPNN algorithm matches well with the calibration point in the pressure range 0~1.1 MPa and temperature range −40 °C~60 °C, with a significant reduction of error. As a result, the algorithm’s maximum absolute error is 0.2228 kPa, and its full-scale error is 0.02% FS. As far as the temperature compensation of the pressure scanner is concerned, the algorithm shows strong adaptability. Simultaneous measurement of multi-channel pressure is achieved while ensuring high accuracy.

However, in proposing a temperature compensation model for a multi-channel pressure scanner, this paper does not strictly consider the zero-point drift of the adiabatic pressure sensor at different pressure points and temperatures. This will also contribute to errors in measurement results in practical applications, so further research is needed to address this issue. In addition, the measured value of the pressure scanning valve’s internal temperature sensor output is ideal by default in this paper. In fact, the non-linear error of the temperature sensor affects the compensation result, which requires further study.

## Figures and Tables

**Figure 1 micromachines-13-01351-f001:**
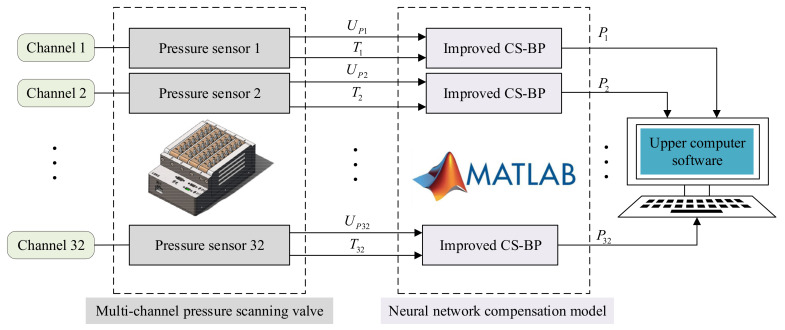
Temperature compensation schematic diagram of the multi-channel pressure scanner.

**Figure 2 micromachines-13-01351-f002:**
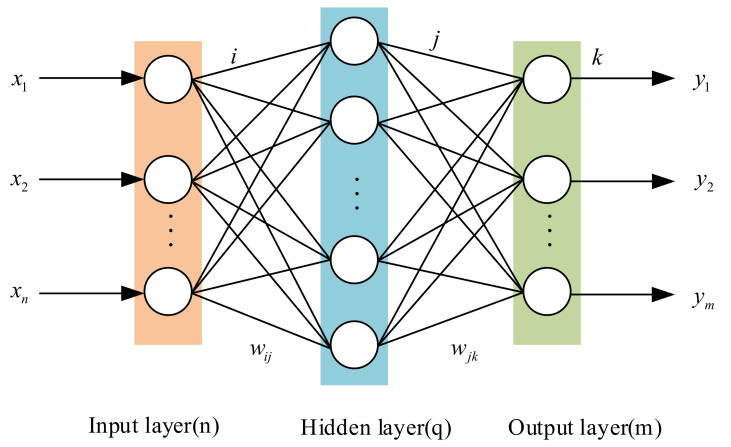
Structure diagram of BP neural network.

**Figure 3 micromachines-13-01351-f003:**
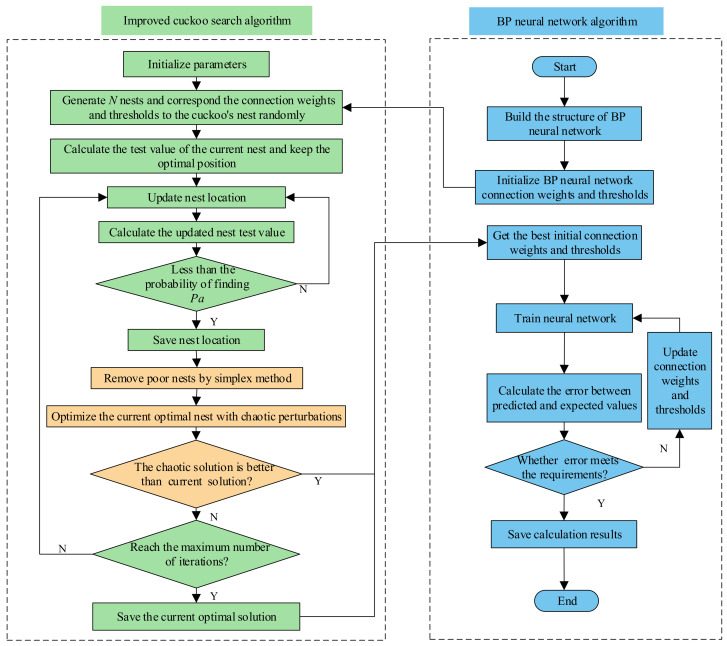
Improved CS-BP algorithm flow chart.

**Figure 4 micromachines-13-01351-f004:**
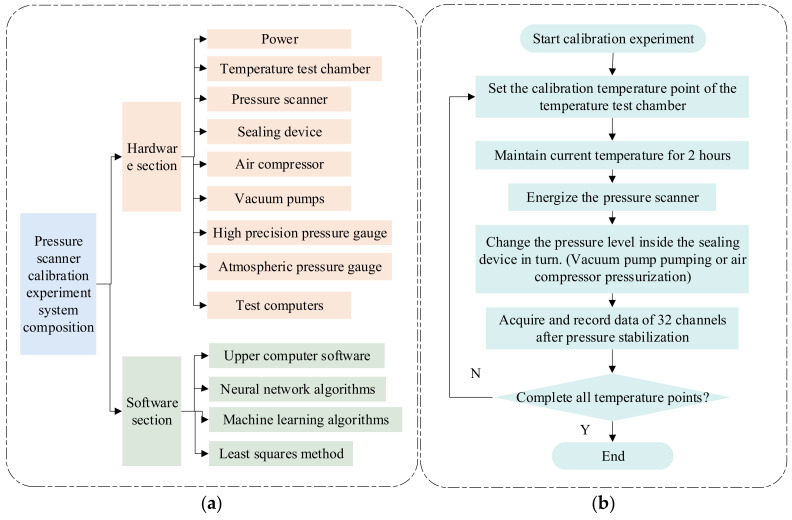
(**a**) Multi-channel pressure scanner calibration experiment system composition diagram. (**b**) The procedure of the calibration.

**Figure 5 micromachines-13-01351-f005:**
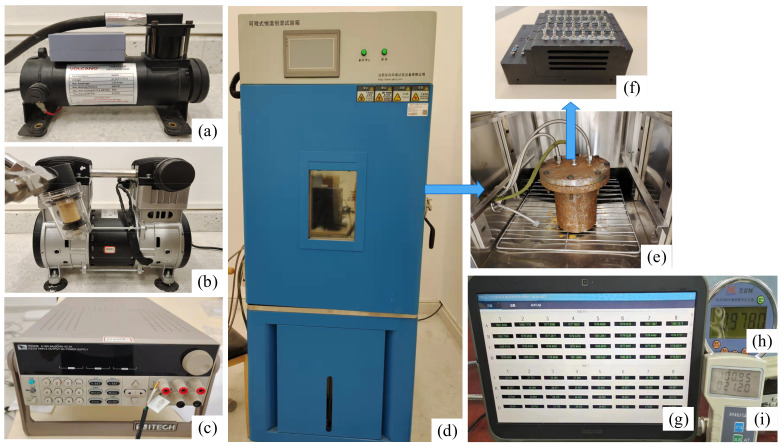
Calibration experiment system of multi-channel pressure scanner: (**a**) BM400 air com-pressor, (**b**) SY-3000H vacuum pump, (**c**) ITECH IT6332B DC regulated power supply, (**d**) thermostatically and humidity-controlled test chamber, (**e**) sealing device, (**f**) 32-channel pressure scanner, (**g**) test computer, (**h**) ALKC400H precision digital pressure gauge, (**i**) digital temperature–atmosphere pressure gauge.

**Figure 6 micromachines-13-01351-f006:**
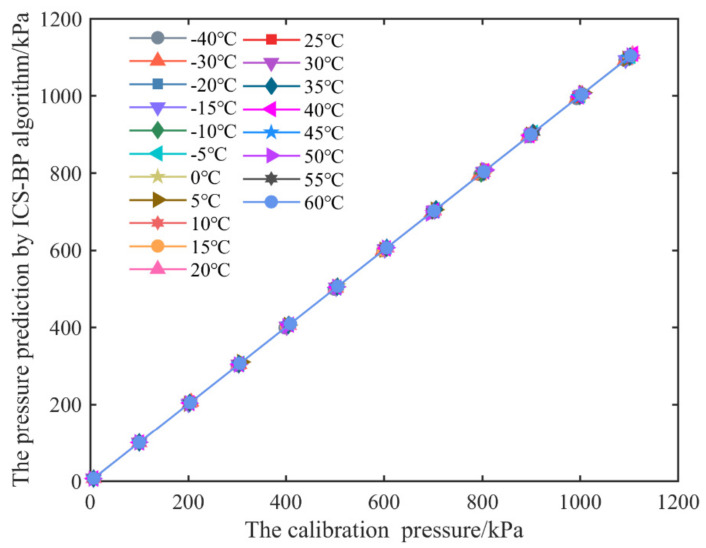
Relation curve between calibration pressure and predicted pressure by ICS-BP algorithm.

**Figure 7 micromachines-13-01351-f007:**
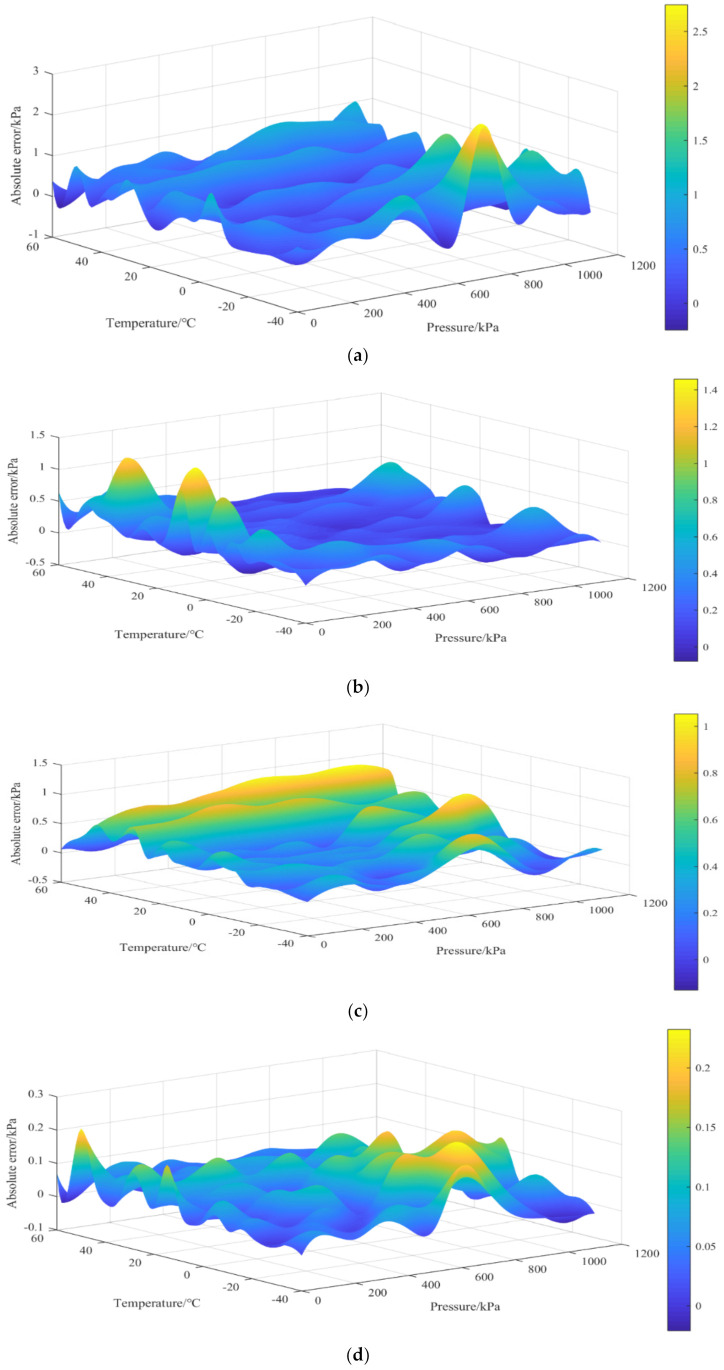
Experimental results: (**a**) the least−squares method, (**b**) RBF neural network (**c**), BP neural network, (**d**) improved CS-BPNN.

**Figure 8 micromachines-13-01351-f008:**
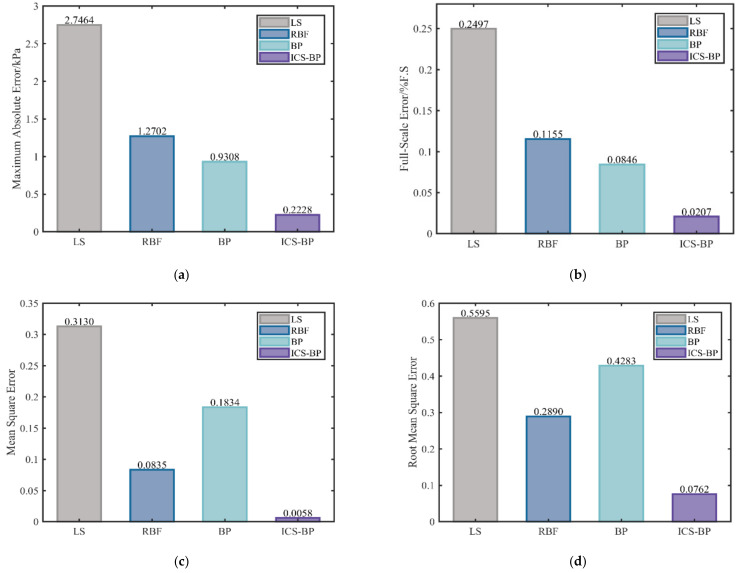
Experimental results: (**a**) maximum absolute error, (**b**) full-scale error, (**c**) mean square error, (**d**) root mean square error.

**Figure 9 micromachines-13-01351-f009:**
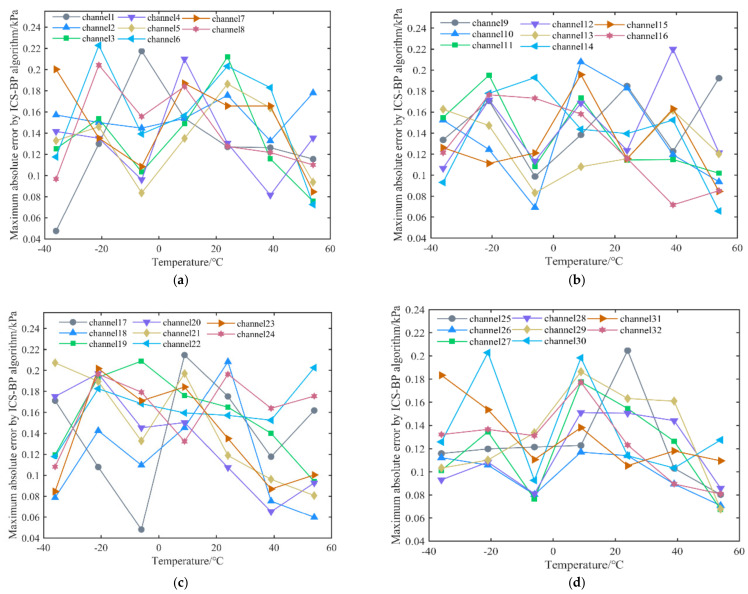
Experimental results: (**a**) Compensation test results for channels 1−8. (**b**) Compensation test results for channels 9−16. (**c**) Compensation test results for channels 17−24. (**d**) Compensation test results for channels 25−32.

**Table 1 micromachines-13-01351-t001:** Calibration data at full temperature.

$T_{1}/{}^{\circ}C$	$U_{P1}/V$											
	0	100 kPa	200 kPa	300 kPa	400 kPa	500 kPa	600 kPa	700 kPa	800 kPa	900 kPa	1000 kPa	1100 kPa
−40	−0.6544	0.0721	0.8995	1.6354	2.3938	3.1634	3.9241	4.6678	5.4448	6.2075	6.9702	7.7329
−30	−0.6446	0.0675	0.8847	1.6304	2.3950	3.1272	3.8396	4.5769	5.2963	6.0775	6.8224	7.5674
−20	−0.6363	0.0595	0.8587	1.6064	2.3264	3.0791	3.7968	4.5311	5.2971	6.0223	6.7428	7.4892
−15	−0.6305	0.0592	0.8132	1.5766	2.3069	3.0439	3.7715	4.5089	5.2366	5.9463	6.7028	7.3531
−10	−0.6243	0.0602	0.8233	1.5591	2.3106	3.0358	3.7664	4.4836	5.1590	5.9089	6.6099	7.3398
−5	−0.6150	0.0653	0.8156	1.5568	2.2832	2.9770	3.7009	4.4195	5.1359	5.8534	6.5319	7.2584
0	−0.6143	0.0595	0.8020	1.5332	2.2562	2.9790	3.6479	4.3764	5.0720	5.7336	6.4786	7.2054
5	−0.6098	0.0582	0.7831	1.5535	2.2338	2.9326	3.6121	4.3407	5.0459	5.7050	6.4591	7.0797
10	−0.6043	0.0533	0.7867	1.4924	2.2027	2.8981	3.5666	4.2766	4.9530	5.6362	6.3259	7.0361
15	−0.5919	0.0535	0.7757	1.4783	2.1566	2.8808	3.5260	4.2213	4.9645	5.5954	6.2701	6.9652
20	−0.5884	0.0575	0.7715	1.4520	2.1655	2.8164	3.5328	4.1861	4.8657	5.5198	6.2383	6.9312
25	−0.5780	0.0616	0.7847	1.4486	2.1301	2.8145	3.4841	4.1436	4.8074	5.4984	6.1124	6.8323
30	−0.5687	0.0678	0.7655	1.4440	2.1053	2.7883	3.4591	4.1319	4.7978	5.4717	6.1020	6.7908
35	−0.5663	0.0620	0.7555	1.4275	2.1018	2.7673	3.4373	4.0972	4.7404	5.4039	6.0393	6.7024
40	−0.5594	0.0610	0.7488	1.4143	2.0923	2.7319	3.3950	4.0350	4.7034	5.3080	5.9640	6.6825
45	−0.5496	0.0646	0.7439	1.4003	2.0722	2.7120	3.3745	3.9868	4.6585	5.2856	5.9044	6.5588
50	−0.5545	0.0643	0.7241	1.3731	2.0168	2.6703	3.3235	3.8903	4.6142	5.1811	5.8767	6.5064
55	−0.5432	0.0560	0.7141	1.3700	2.0314	2.6536	3.2769	3.8782	4.5352	5.1928	5.8147	6.4163
60	−0.5355	0.0610	0.7232	1.3682	2.0208	2.6323	3.2610	3.8646	4.5009	5.1045	5.7520	6.3763

## Data Availability

The data presented in this study are available on request from the corresponding author.

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
