# Peer review of "Research on Temperature Compensation of Multi-Channel Pressure Scanner Based on an Improved Cuckoo Search Optimizing a BP Neural Network"

_micromachines, 2022, doi:10.3390/mi13081351_

Round 1

Reviewer 1 Report

The manuscript "Research on temperature compensation of multi-channel pressure scanner based on improved CS-BPNN" presents an improved tempeature compensation algorithm for multi-channel pressure scanners. Although the proposed method is rather interesting, it is required that the authors perform various mandatory modifications to their manuscript before it can be accepted for publication.

In the Abstract section, line 16, please correct "optimizes" to "optimize". Moreover, "FS" should be explained.

In the Introduction section or also in section 2, the authors should justify that their subject is both relevant to the scope of the journal and MEMS technology as well. Thus, they should add at least 2 paragraphs about the relation between the pressure sensor and MEMS and add some relevant papers. Moreover, they should add more papers about compensation methods used in the relevant literature, in order to justify their novelty.

In section 2, pages 4-5 the authors present the basic equations describing the BPNN model. However, as they state that the transfer functions are tansig and purelin, whereas the training method is trainlm, they should also provide the necessary formulas regarding them. In subsection 2.1.2, lines 155-156, the sentence "Matching pa only with complex problems in practice" is incomplete and should be corrected. 

The authors should present some more details about cuckoo search algorithm and explain in more details the step c) of the proposed algorithm, mentioned in page 5.

Regarding the algorithm presented in the flowchart of Figure 3, it seems that the optimization algorithm is only used to determine the initial weights of the BPNN whereas the final weights are determine by the classic approach. If this is true, this contradicts the statement made by the authors in lines 194-195, that the parameters of the most suitable BPNN are found by the cuckoo algorithm. Which criteria are used to determine the optimum initial weights?  

Regarding the "calibration experiment system" described in subsection 2.2, a simple flowchart indicating the components of the system and the procedure of the calibration should be added.

The devices and test stand used for the test experiments should be displayed in an appropriate figure. In subsection 3.1, the authors should explain which signals are measured with the 32 channels. 

In subsection 3.2, the authors should state which was the percentage of the total samples used for training and test.

In subsection 3.4, the authors should mention and comment on which sensor or sensors exhibited higher errors.

Author Response

Thanks to the reviewers, the revised comments are available in pdf.

Reviewer 2 Report

The article contains information technical and innovative. The problem addressed is current and has technical relevance, which makes it significant.  The paper is well organized and convincing. The experimental methodology is described comprehensively. Interpretations and conclusions are justified by the results. The paper is well organized and convincing. 

My recommendations are:

a)Acronyms should be avoided in the title of the article.

b)Spell out each acronym the first time used in the body of the paper. Spell out acronyms in the Abstract.

c) The abstract should clarify what is exactly proposed (the technical contribution) and be rewritten to be more meaningful;

d) Rewrite the precise sentence "The figure below illustrates the basic structure of a practical layer BP neural network." I believe it's figure 2

e) You apply the simplex method. Wouldn't an interior point algorithm be better?

f) More extensive simulations and more figures are needed
g) Discuss the future plans with respect to the research state of progress and its limitations.

Author Response

(The authors gave the same response as above.)

Round 2

Reviewer 1 Report

The authors have performed most of the required modifications to their manuscript. Thus it can be recommended for publication. Before submitting the final manuscript, in the paragraph between lines 88-103 the authors should avoid referring to papers as "the literature [21]" but they should appropriately mention the names of the authors e.g. "In the work of Zhang et al. [21]". Moreover, "Levenbery-Marquardt" should be corrected to "Levenberg-Marquardt".

Reviewer 2 Report

Considering that the authors presented satisfactory answers to the required questions, this reviewer is in favour of its publication.